# Crown Width Extraction of *Metasequoia glyptostroboides* Using Improved YOLOv7 Based on UAV Images

Chen Dong [1,2,3], Chongyuan Cai [1,2,3], Sheng Chen [4], Hao Xu [5], Laibang Yang [6], Jingyong Ji [7], Siqi Huang [8], I-Kuai Hung [9], Yuhui Weng [9] and Xiongwei Lou [1,2,3,*]

1   College of Mathematics and Computer Science, Zhejiang A & F University, Hangzhou 311300, China; dongchen@zafu.edu.cn (C.D.); 2020611011003@stu.zafu.edu.cn (C.C.)
2   Key Laboratory of State Forestry and Grassland Administration on Forestry Sensing Technology and Intelligent Equipment, Zhejiang A & F University, Hangzhou 311300, China
3   Key Laboratory of Forestry Intelligent Monitoring and Information Technology Research of Zhejiang Province, Zhejiang A & F University, Hangzhou 311300, China
4   Center for Forest Resource Monitoring of Zhejiang Province, Hangzhou 310000, China
5   Zhejiang Forestry Bureau, Hangzhou 310000, China
6   Hangzhou Ganzhi Technology Co., Ltd., Hangzhou 311300, China
7   Longquan Forestry Bureau, Longquan 323700, China
8   Longquan Urban Forestry Workstation, Longquan 323700, China
9   College of Forestry and Agriculture, Stephen F. Austin State University, Nacogdoches, TX 75962, USA
*   Correspondence: lxw@zafu.edu.cn; Tel.: +86-135-8822-8755

**Abstract:** With the progress of computer vision and the development of unmanned aerial vehicles (UAVs), UAVs have been widely used in forest resource investigation and tree feature extraction. In the field of crown width measurement, the use of traditional manual measurement methods is time-consuming and costly and affects factors such as terrain and weather. Although the crown width extraction method based on the segmentation of UAV images that have recently risen in popularity extracts a large amount of information, it consumes long amounts of time for dataset establishment and segmentation. This paper proposes an improved YOLOv7 model designed to precisely extract the crown width of *Metasequoia glyptostroboides*. This species is distinguished by its well-developed terminal buds and distinct central trunk morphology. Taking the *M. glyptostroboides* forest in the Qingshan Lake National Forest Park in Lin'an District, Hangzhou City, Zhejiang Province, China, as the target sample plot, YOLOv7 was improved using the simple, parameter-free attention model (SimAM) attention and SIoU modules. The SimAM attention module was experimentally proved capable of reducing the attention to other irrelevant information in the training process and improving the model's accuracy. The SIoU module can improve the tightness between the detection frame and the edge of the target crown during the detection process and effectively enhance the accuracy of crown width measurement. The experimental results reveal that the improved model achieves 94.34% mAP@0.5 in the task of crown detection, which is 5% higher than that achieved by the original model. In crown width measurement, the $R^2$ of the improved model reaches 0.837, which is 0.151 higher than that of the original model, thus verifying the effectiveness of the improved algorithm.

**Keywords:** unmanned aerial vehicle; forest resources; crown width; YOLOv7; attention module

## 1. Introduction

Forests, an important part of the terrestrial biosphere, play a great role in the carbon cycle of the terrestrial biosphere [1], not only in terms of the economic benefits but also in ecosystem services related to forest landscapes [2,3]. As an important part of forests, trees play a vital role in maintaining terrestrial ecological balance, regulating the climate, maintaining soil and water, and sealing and fixing carbon [4–6]. The relic plant *Metasequoia glyptostroboides* is a unique gymnosperm in China and a national first-class protected

plant. Although *M. glyptostroboides* is widely cultivated around the world, some problems remain in its cultivation, such as poor natural regeneration and racial decline, and thus the protection of the *M. glyptostroboides* population is facing a severe challenge [7]. As the top layer of vertical distribution, the crown is the main location for the photosynthesis and transpiration of trees. The crown size and the leaf characteristics represent the growth status of trees [8–10]. From an ecological point of view, the crown directly affects the distribution of undergrowth plants and animal communities. It plays a role in regulating light intensity, precipitation distribution, and nutrient circulation [11]. Crown width, which is the arithmetic average of the widths of trees in the north-south and east-west directions [12], is usually used to express the specifications of trees. Crown width is an effective indicator for measuring the health status of trees and an important factor in evaluating related ecological services [13]. In forestry management and forest resources investigation, crown width is one of the important characteristic factors of the crown structure and an important variable in predicting the crown surface area, the canopy density, and the tree biomass [14]. Crown width is an important feature of any tree growing in an open environment because it is related to many different forest management factors, such as photosynthesis, trunk diameter, carbon, water and energy exchange, tree competition, tree health, and growth efficiency [15–18]. The dynamic crown width monitoring of the *M. glyptostroboides* population helps researchers grasp the growth of *M. glyptostroboides* in real-time, making the collection of *M. glyptostroboides* crown information significant.

The traditional forest resources investigation is mainly based on field investigation. In crown width measurement, the projection and vertical aiming methods are the traditional field manual measurement methods [19]. Although the vertical aiming method is faster than the projection method, it is not as accurate, while the projection method is time-consuming and has low measurement efficiency. Both methods are restricted by weather, the environment, topography, and other factors, and limited samples can be collected, which may lead to a certain degree of deviation in the results [20]. Therefore, an accurate crown width estimation model must be established by using a large number of samples. Currently, crown estimation is commonly performed to obtain the estimated crown width value by inputting the forest and tree characteristics into the model as certain or random information [21]. The mathematical crown width estimation model has gone through the regression process, from the simple least square method to nonlinear mixed effect modeling [22,23]. However, with the development of science and technology, developing methods to measure the crown width with new technology has become a hot topic in recent years. Meanwhile, the resolution of remote sensing images has been greatly improved. For example, the spatial resolution of QuickBird satellite images has reached 0.61 m. On remote sensing images of such levels, the crown width of a single tree with a relatively small canopy can be directly estimated [24]. However, in the stand with high canopy density, individual trees are difficult to identify in the image, making the crown width difficult to estimate. The appearance of unmanned aerial vehicles (UAVs) has effectively solved the insufficient resolution of remote sensing images. In forestry, an increasing number of methods for detecting the tree crown and measuring the tree crown width using UAVs equipped with lidar and high-definition cameras have been developed under the cooperation of deep learning technology. For example, Weinstein et al. proposed a method of identifying the tree crown on RGB images using a semi-supervised learning detection network, achieving an average detection rate of 82% [25]. Taking spruce as the study object, Emin et al. used four target detection models to identify the crown of spruce at low, medium, and high densities, among which the recognition accuracy of the Faster-RCNN model reached 96.36%, 96.32%, and 95.54% at three densities, respectively [26]. Chen et al. used unmanned airborne LiDAR to obtain sample data and segmented a single crown from the sample through the PointNet deep learning framework [27]. Panagiotidis et al. reconstructed the 3D structure of a sample plot using UAVs and computer vision technology and estimated the diameter of the crown width by using inverse watershed segmentation technology with the help of a geographic information system [28]. With

the visible light image obtained by UAVs as the data source, Ye et al. used the U2-Net deep learning model to segment the olive tree and extract the crown information of the sample plot [29]. Wu et al. used a UAV to acquire remote images of an orchard, identified apple trees in the images by a target detection model, and then segmented the identified apple tree canopies using a U-Net model and calculated the relevant canopy information from the segmentation results, ultimately achieving 92% accuracy in measuring the canopy width [30].

Segmentation technology can extract more information about the crown; however, it is complicated to use in terms of dataset establishment and needs outlining along the crown edge. The training duration of segmentation models in the same experiment environment is typically longer than that of target detection models. Wu et al.'s method is divided into detection and segmentation. In order to simplify this process, Lou et al. proposed a loblolly pine crown-width measurement method based on object detection algorithms. They used three object detection models to conduct experiments on loblolly pine forests for two different years, which proved that the object detection model is very convenient and fast in crown detection and crown width measurement [31]. Both Loblolly Pine (*Pinus taeda*) and *M. glyptostroboides* are classified as tall trees. However, the architecture of their branch systems is different. The auxiliary branches of mature loblolly pine trees are relatively scattered among the main branches, potentially causing visual confusion between a mature loblolly pine and multiple trees. By contrast, the auxiliary branches of *M. glyptostroboides* grow close to the main branches, creating a more compact branch structure. In high-density planting, two *M. glyptostroboides* grown at adjacent distances may appear as a single tree due to their closely grown branches. In target detection in a dense *M. glyptostroboides* forest, utilizing a two-stage detector with a candidate area generation stage may lead to the erroneous grouping of nearby *M. glyptostroboides* into a single entity. Conversely, a single-stage detector directly generating target class probabilities and location coordinates is better suited for such a challenging scenario. Therefore, this study uses the single-target detection model to explore the crown width measurement for *M. glyptostroboides*. To achieve this goal, the latest single-stage target detection model, YOLOv7 [32], was improved to enable crown identification and the accurate measurement of crown widths. The improved YOLOv7 model has been experimentally proven capable of coping with small-crown-width and high-density stand environments more effectively than other deep learning models.

## 2. Materials

### 2.1. Overview of the Study Area

The study area is located in the Qingshan Lake National Forest Park in Lin'an District, Hangzhou City, Zhejiang Province, China. Situated at the end of Tianmu Mountain, the study area belongs to the subtropical monsoon climate, with a warm and humid climate and abundant rainfall and sunlight. The average annual precipitation is 1613.9 mm, with 158 days of precipitation and 237 days of a frost-free period. The Qingshan Lake Park covers a total area of 984 hectares, with both natural and artificial vegetation. The natural vegetation includes coniferous and broad-leaved mixed forests, shrubs, and aquatic plants, and the artificial vegetation includes economic forests, tea gardens, and *M. glyptostroboides* forests. *M. glyptostroboides* is the main tree species in the sample plot of the study area. The sample plot is composed of pure trees with flat terrain and complex background, and the background color is close to the color of the tree canopy, which is suitable for verifying the effect of the target detection model in terms of crown identification and crown width extraction. The study area is displayed in Figure 1.

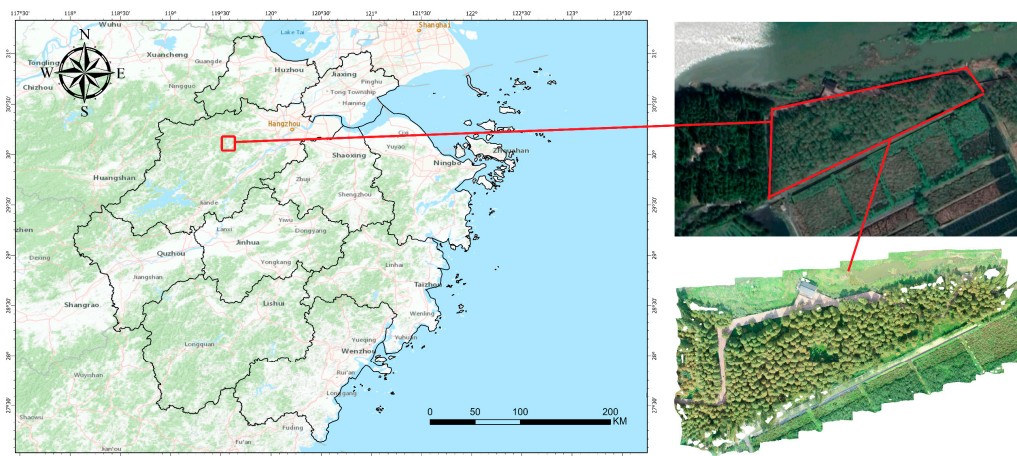

**Figure 1.** Areal map of the sample plot.

## 2.2. Data Acquisition

In this study, UAVs were used as data acquisition equipment, Pix4Dcapture was used as the UAV flight control software, and the orthograph of the sample plot of the study area was generated using Agisoft PhotoScan. The UAVs used in this study are the Phantom4 Pro V2.0 in DJI, equipped with a 1-inch 20-megapixel image sensor and a maximum flight time of 30 min. To reduce the shadow of the UAVs in the imaging process and ensure their sufficient imaging ray, data collection should be carried out within a period of sunny and windless weather with stable light intensity. In this experiment, the flight height of the UAVs was set to 50 m, the camera angle was 90° vertically downward, the overlapping image ratio was set to 90%, and the flight speed of the UAVs was set to 27 km/h. The flight route and real-time captured images of the UAVs are exhibited in Figure 2.

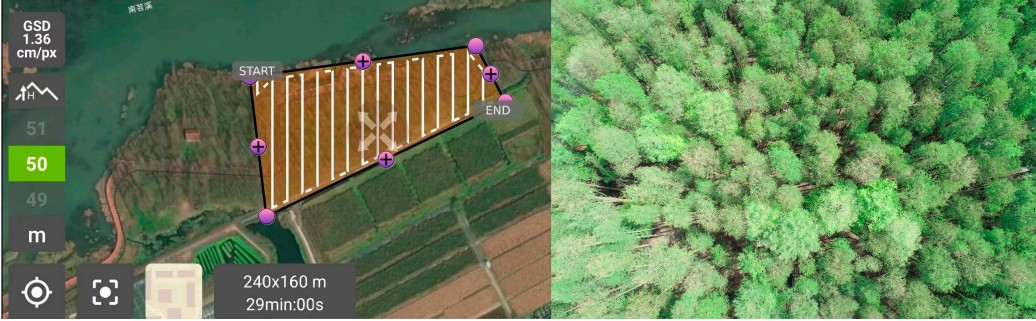

**Figure 2.** Aerial route planning map of UAVs and real-time captured image of the sample plot.

The original format of the captured image data was JPEG, which contained accurate GPS coordinates and POS data. The original image data were imported into Agisoft PhotoScan to generate a high-resolution orthograph. The size of the obtained orthograph of the study area was 24,425 × 15,376 pixels, with a resolution of 0.01155 m per pixel. The generated orthograph is shown in Figure 3.

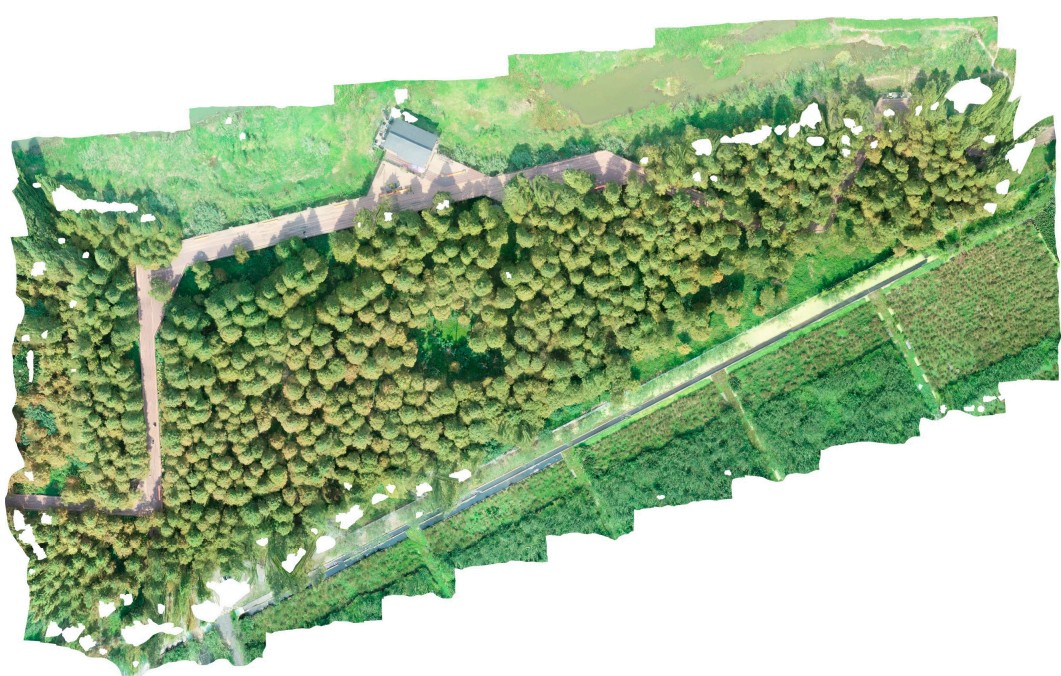

**Figure 3.** Orthograph of sample plot in *M. glyptostroboides* forest.

In general, the crown width measurement should be based on the field measurement. Given that the sample plot has a fence outside and *M. glyptostroboides* grow in the shallow waters of the lake, field measurements near the trees are impossible due to objective constraints. Therefore, the measurement method should be adjusted according to the actual conditions. The resolution of each pixel in the orthograph was known in this experiment, and thus the actual crown length and width could be converted by the number of pixels in the map, and the converted average value of the length and the width is the actual crown width.

### 2.3. Dataset Establishment

A part of the entire orthograph was clipped and used as a sample, and the unclipped section was used as the test set to evaluate the model's effectiveness. The size of each sample picture was $500 \times 500$ pixels, and each picture contained several *M. glyptostroboides* crowns, providing a total of 256 pictures. These pictures were manually labeled using the LabelImg tool, and from each labeled picture, a corresponding XML file was generated and saved in the PASCAL VOC format [33]. The file contained information, such as the saving path, target category, and frame coordinates of the picture, and the labeled picture is shown in Figure 4.

### 2.4. Mixup Data Enhancement

Under normal circumstances, the target detection model based on convolutional neural networks requires a large amount of data to achieve good results. In the field of target detection and classification, scholars generally believe that the performance of the model is positively correlated with the number of samples. However, difficulties are often unavoidable in the collection of some special types of data. Consequently, a small amount of data is available in the training stage of the model, which may result in the over-fitting of the model and affect the ultimate target detection performance of the model.

In view of this phenomenon, the common methods in the industry at present include early stop, freezing training, transfer learning, regularization, and data enhancement. Data enhancement expands the training set, thus enhancing the robustness of the model, improving the generalization ability of the model, and reducing the over-fitting phenomenon of the model. For the forest with a single tree species, the extraction of the edge texture

features of the crown should be strengthened in the training process to increase the accuracy of identification; the trees in the forest are often staggered and overlapped, and the light distribution inside the crown is uneven. To simulate the tree interleaving situation in the dataset, the dataset was enhanced in this experiment using the mixup method. Mixup is a data enhancement method published on ICLR in 2018 [34]. The core idea is to randomly select two images from each batch and fuse them in a certain proportion to generate a new image. Meanwhile, the one-hot codes corresponding to the images are also multiplied in the same proportion to construct a new image. In this study, the new image was added to the original dataset to form a new dataset, and the new data image generated by fusion is shown in Figure 5.

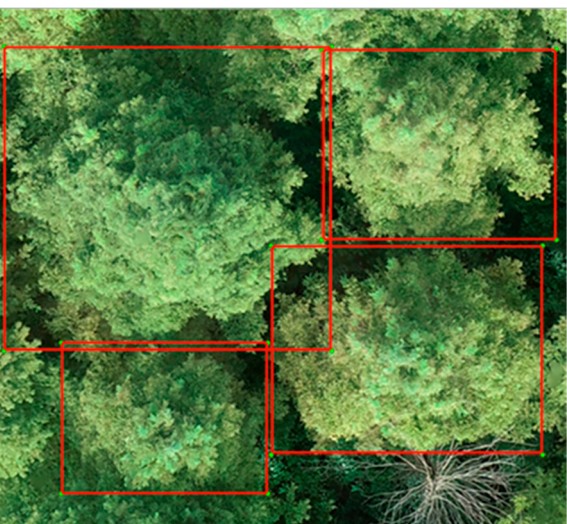

**Figure 4.** Crowns labeled by LabelImg tool.

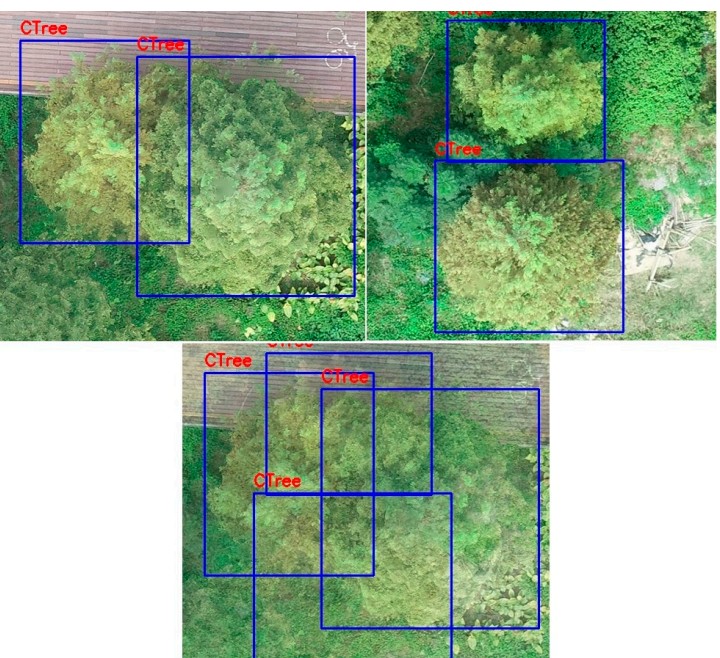

**Figure 5.** New image enhanced by mixup data. The upper two images are original images; the lower image is a new image generated by fusion. To facilitate distinguishment, the crown of the tree is marked and displayed with a blue frame.

## 3. Method

### 3.1. Description of YOLOv7 Algorithm

The target detection model is divided into two- and single-stage detectors. The representative two-stage detector is Faster-RCNN, and the representative single-stage detectors are the SSD and YOLO series. In this study, the single-stage detector YOLOv7 was improved, and a YOLOv7 crown detection and crown width measurement algorithm that combines the attention mechanism and the brand-new loss function was proposed. Through this algorithm, the forestry staff could quickly count the number of single trees and measure the crown width in the sample plot area, which is conducive to the rapid collection and management of forestry resource information.

In view of the different growth years of *M. glyptostroboides* in the study plot, some *M. glyptostroboides* trees had small crowns, and the overall study area was large. The latest YOLOv7 model was selected to improve detection efficiency. YOLOv7 is a single-stage detector with the latest architecture in the YOLO series. YOLOv7 exhibits faster detection speeds and higher accuracy than the previous YOLO series detectors and proposes an efficient network architecture, which saves on reasoning costs and makes model training efficient without accuracy reduction. At the same FPS, YOLOv7 is 120% faster than YOLOv5. The test results on the MS COCO dataset are better than those of the YOLOv5 detector. Figures 6 and 7 show the network structure and basic network module of YOLOv7, respectively.

The structure chart demonstrates that YOLOv7 consists of four parts: the input, backbone, head, and prediction modules. The YOLOv7 network preprocessed the image first, adjusted the image size to $640 \times 640 \times 3$, and transmitted the processed image to the backbone network. The backbone network layer comprised several CBS, ELAN, and MP modules. Through these three modules, the length and width of the feature map were alternately reduced by 1/2, and the number of output channels was increased to twice the number of input channels. As shown in Figure 7, the CBS modules performed the Conv+BN+SiLu operation on the input feature map, where *K* below the CBS module denotes the size of the convolution kernel, and *S* denotes the stride. The activation function of YOLOv7 is the same as that of YOLOv5. Then, SiLu was used to extract the image features of different scales, and the design structure of ELAN was maintained. By guiding different groups of computing blocks to learn diverse features, the learning ability of the network and the model recognition accuracy were continuously improved without the original gradient path. The MP module added the MaxPool layer to the original convolution layer to form two branches, where the upper branch reduced the length and width of the feature map by half through the MaxPool operation, and the number of channels was halved through convolution. In the lower branch, the number of channels was also halved through convolution. Finally, the features extracted from the upper and lower branches were fused through the Concat operation, which improved the feature extraction ability of the network. The head module has a path aggregation feature pyramid network structure. By introducing a bottom-up path, the information at the bottom layer could be easily transmitted to the top layer, thus achieving the purpose of feature fusion at different layers. The prediction module adjusted the number of image channels for three different scales of the output of the features by H-ELAN through the RepConv structure. Finally, $1 \times 1$ convolution was applied to the model prediction and confidence judgment. The advantage of YOLOv7 is that it transforms the multi-branch training model into a high-speed single-branch reasoning model. The final deployed model retains the high accuracy of the multi-branch model and the high efficiency of model training. Finally, YOLOv7 exhibits good recognition accuracy while preserving the advantage, namely, the fast recognition speed of the single-stage detector.

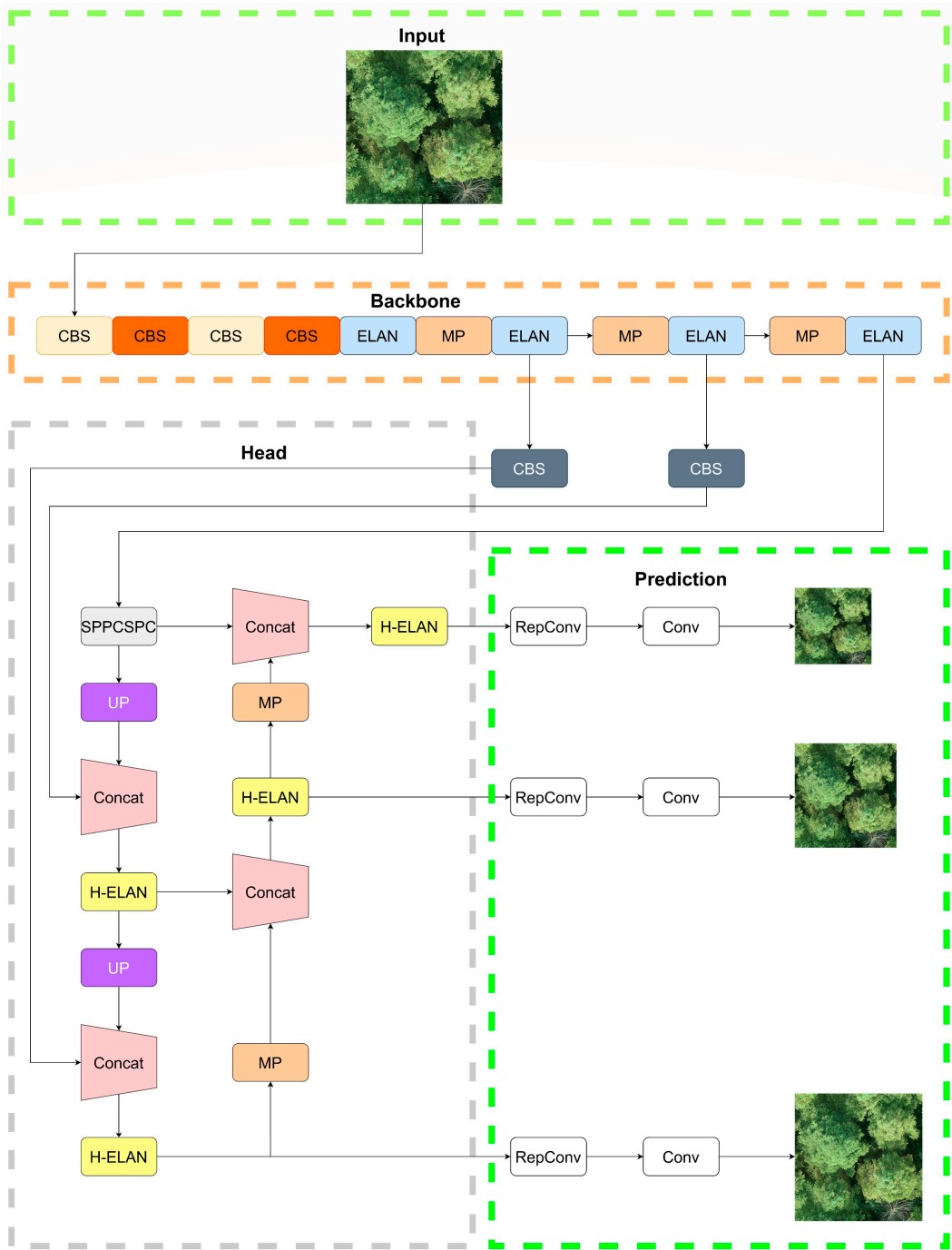

**Figure 6.** YOLOv7 network architecture.

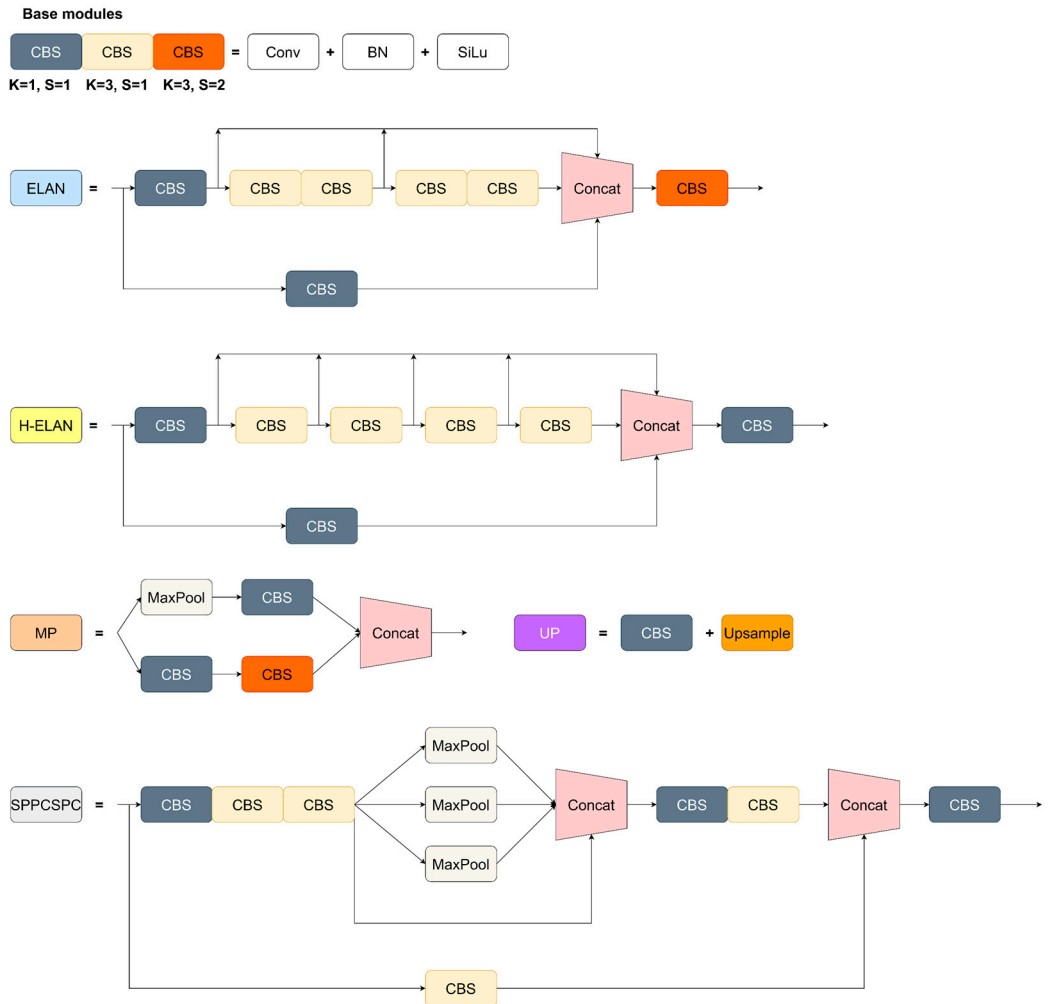

**Figure 7.** YOLO V7 comprises six basic modules, namely CBS, ELAN, H-ELAN, MP, UP, and SPPCSPC.

### 3.2. SimAM Attention Mechanism

In a forest environment, the light intensity of different blocks is often different due to tree interleaving and the change in the solar radiation angle. Especially in the irregularly planted forest area, the shadow and light characteristics vary greatly from block to block due to the scattered distribution of trees, which increases the irrelevant information, which is worthy of attention in the model training process as it results in the larger size of the trained model and will lead to a decline in recognition accuracy. To reduce the interference of such factors in the forest environment, the simple, parameter-free attention model's (SimAM) nonparametric attention module was introduced into the YOLOv7 model in this experiment, enabling the comprehensive and efficient evaluation of the feature weights without introducing additional parameters [35].

The SimAM attention module, which was proposed by Yang et al. in 2021, can infer the three-dimensional attention weight for the feature map in a layer without increasing the original network parameters. On the basis of the theory of neurosciences, the proposal is to optimize the energy function to discover the importance of each neuron, thus enhancing the effective extraction of important features, effectively suppressing the interference of unimportant features, preventing excessive work in structural optimization, and accelerating the calculation of the attention weight. The entire process can be expressed by the following formula:

$$e_t^* = \frac{4(\hat{\sigma}^2 + \lambda)}{(t - \hat{\mu})^2 + 2\hat{\sigma}^2 + 2\lambda} \tag{1}$$

$$\widetilde{X} = sigmoid\left(\frac{1}{E}\right) \odot X \tag{2}$$

According to Equation (1), SimAM evaluates individual neurons within a neural network by establishing an energy function based on their linear separability. Where $\hat{\mu} = \frac{1}{M}\sum_{i=1}^{M} x_i$, $\hat{\sigma}^2 = \frac{1}{M}\sum_{i=1}^{M}(x_i - \hat{\mu})^2$, $t$ represents the target neuron, $x$ is the neighboring neuron, and $\lambda$ is the hyperparameter. Equation (1) indicates the lower energy $e_t^*$; and the neuron $t$ is more distinctive from the surrounding neurons and more important for visual processing. In Equation (2), $X$ is the neuron in a single channel with input features, $E$ stands for the number of all cross-channels and spatial dimension groups, and *sigmoid* represents the activation function. $E$ groups all the $e_t^*$'s across the channel and spatial dimensions, *sigmoid* is added to restrict the value that was too large in $E$, and $\widetilde{X}$ denotes the neuron in a single channel with output features.

The SimAM structure chart is displayed in Figure 8.

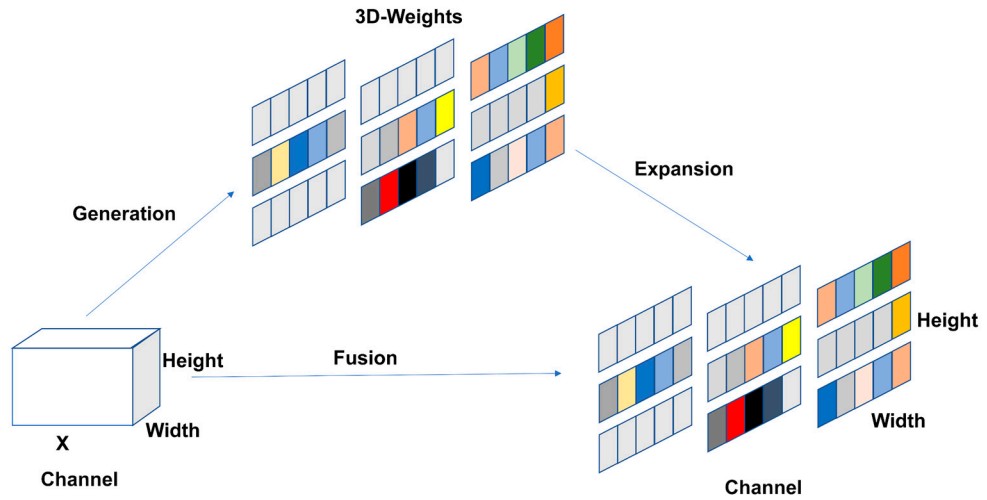

**Figure 8.** SimAM structure chart.

### 3.3. C3 Lightweight Module

In the YOLOv5 model, the authors of the YOLO series transformed the BottleneckCSP module into the C3 module, which is a CSP structure yet differs in the choice of correction units. The C3 module contains three standard convolutional layers and multiple Bottleneck layer modules. Compared with BottleneckCSP, the C3 module removes the convolution module after experiencing the residual output. It replaces the activation function in the standard convolution module after Concat with SiLU, which is the main module for learning the residual features. Its structure is divided into two branches: one only goes through the standard convolution module, and the other goes through the stacked convolution module. Finally, the two branches are combined into one through Concat operation to combine the two branches into one, which can speed up the recognition speed of the model and make the model structure more lightweight while ensuring the feature extraction capability. Figure 9 shows the structure of the C3 module.

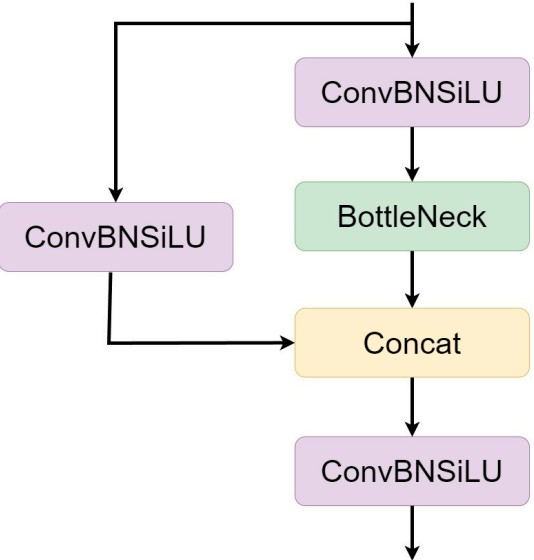

**Figure 9.** Schematic diagram of the structure of the C3 module.

*3.4. SIoU*

In the target detection task, the accuracy of frame recognition depends largely on the loss function definition. The traditional target detection loss function relies on the aggregation degree of regression between the predicted and real frames, such as the aspect ratio and overlapping area between the two; however, their direction matching has not been considered. Influenced by light and topography, trees are often not round in crown shape but present an irregular oval shape. In crown width measurement, the distance between the edges of the predicted and real frames is more important than the overlapping area between the predicted and real frames. In 2022, Zhora et al. proposed a new loss function SIoU, which considers the angle and distance between the predicted and real frames [36]. To extract the crown border effectively and for an accurate crown width measurement, the SIoU was introduced as the loss function in the model training process. The SIoU comprises four parts: the angle, distance, shape, and IoU losses. The definitions of each part are as follows.

(1)　The angle loss is defined as follows:

Figure 10 shows the angle loss, where $C_h$ and $\sigma$ represent the height difference and distance between the central points of the real and predicted frames, respectively.

$$\wedge = 1 - 2 \times \sin^2\left(\arcsin\left(\frac{C_h}{\sigma}\right) - \frac{\pi}{4}\right) = \cos\left(2 \times \left(\arcsin\left(\frac{C_h}{\sigma}\right) - \frac{\pi}{4}\right)\right) \tag{3}$$

(2)　The distance loss is defined as follows:

Figure 11 shows the distance loss, where $\rho_x = \left(\frac{b_{C_x}^{gt} - b_{C_x}}{C_w}\right)^2$, and $\rho_y = \left(\frac{b_{C_y}^{gt} - b_{C_y}}{C_h}\right)^2$, and $\gamma = 2 - \wedge$.

$$\Delta = \sum_{t=x,y} \left(1 - e^{-\gamma \rho_t}\right) \tag{4}$$

(3)　The shape loss is defined as follows:

$$\Omega = \sum_{t=w,h} \left(1 - e^{-\omega_t}\right)^\theta \tag{5}$$

where $\omega_w = \frac{|w - w^{gt}|}{max(w - w^{gt})}$ and $\omega_h = \frac{|h - h^{gt}|}{max(h - h^{gt})}$.

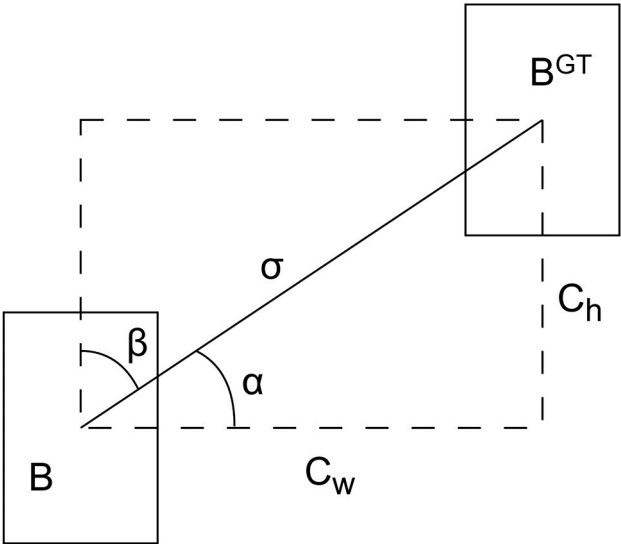

**Figure 10.** Angle loss.

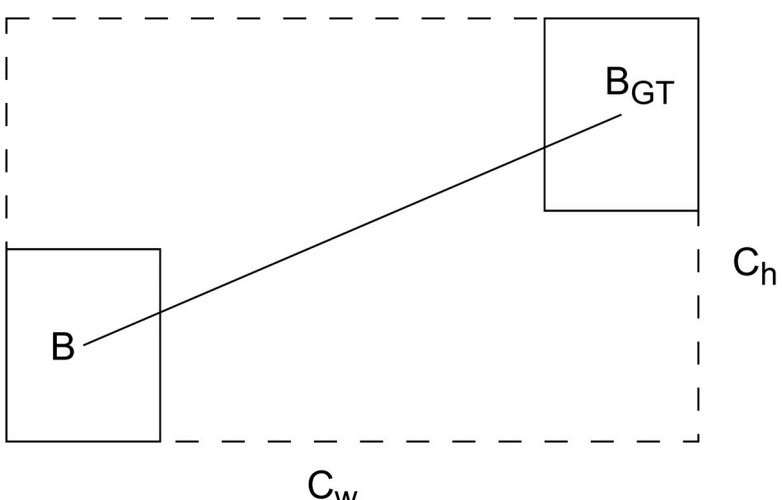

**Figure 11.** Distance loss.

(4)   The IoU loss is defined as follows:

Figure 12 shows the IoU loss, the final loss function is defined as follows:

$$Loss_{SIoU} = 1 - IoU + \frac{\Delta + \Omega}{2} \tag{6}$$

### 3.5. Improved YOLOv7 Network Model

Given that most forest areas are large, considering the practical application, the model should be lightweight on the premises of ensured model accuracy, the identification speed must be accelerated, and the model training time should be shortened. The C3 module is a lightweight module proposed in YOLOv5. Given that YOLOv5 has been widely used in industrial research with its verified practical application, the C3 module within it was also introduced into the YOLOv7 network in this study to achieve a lightweight model without influencing the model accuracy.

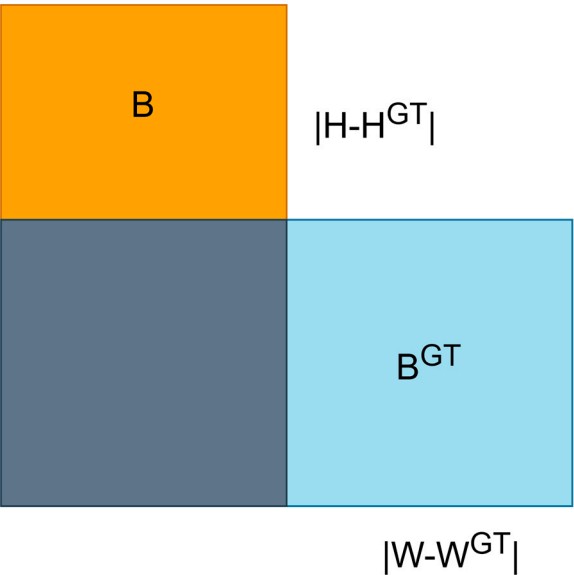

**Figure 12.** IoU loss.

Figure 13 shows the improved YOLOv7 network structure. By replacing the ELAN of the input and output sections in Backbone with C3, the output and input parameters of the model were reduced, and the ELAN in the middle portion was reserved to ensure the model's accuracy. To achieve vector consistency between the output and input sections, the CBS module was added after the two substitutive C3 modules in Backbone.

In the Head portion, the H-ELAN module was replaced as a whole with the C3 module, which further accelerated the model's training speed. The model size was reduced by 10% after model training.

In the first MP module of the head portion, one CBS module was substituted by the SimAM module to enhance the model's ability to extract the crown edge features. Compared with the 3D parametric attention mechanism, SimAM could evaluate the feature weights comprehensively and efficiently without introducing additional learnable parameters, thus enhancing the edge texture features of the crown, weakening the background interference of the complex forest environment, improving the anti-interference ability of the model, and strengthening the robustness of the model.

*3.6. Evaluation Indexes*

The model evaluation was divided into two parts: crown identification and crown width measurement. In the first part, the precision, recall, F1-score, and mAP@0.5 were used as the evaluation indexes. In the second part, bias, RMSE, and $R^2$ were used as the evaluation indexes through a comparison with the true values.

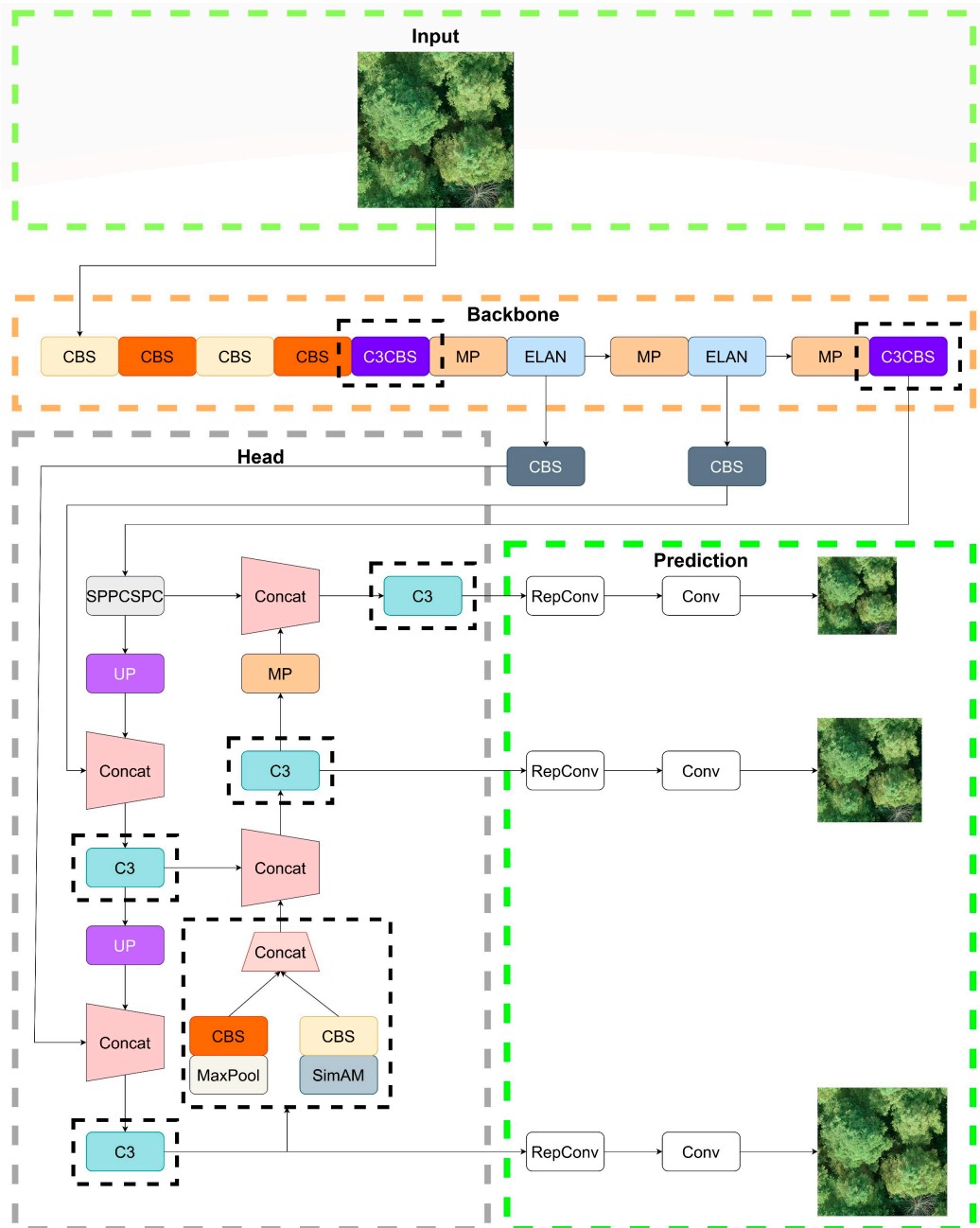

**Figure 13.** Improved YOLOv7 network structure, where the black frames at 7 locations represent the improvements of the YOLOv7 network structure.

Among the four indexes of crown identification, precision represents the proportion of real targets among the positive samples in the model prediction. Recall denotes the proportion of the positive samples predicted by the model in the number of real positive samples. The F1-score, which is also known as the balanced F-score, is defined as the harmonic average of precision and recall, which considers precision and recall. The mAP@0.5 indicates the part in which the prediction accuracy is higher than 50% in the validation set and stands for the average prediction accuracy of the model. These indexes are expressed as follows:

$$\text{Precision} = \frac{TP}{TP + FP} \tag{7}$$

$$\text{Recall} = \frac{TP}{TP + FN} \tag{8}$$

$$F1\text{-}Score = 2 \times \frac{Precision \times Recall}{Precision + Recall} \tag{9}$$

where *TP* is the number of correctly classified positive samples, that is, the number of correctly identified tree crowns; *FP* is the number of wrongly classified positive samples, that is, the number of wrongly identified tree crowns; and *FN* is the number of wrongly classified negative samples.

Among the three indexes of the crown width measurement, bias represents the deviation between the estimated value and the actual value, *RMSE* is used to reflect the discretization degree of data, and $R^2$ is used as the main index to evaluate the accuracy of the crown width prediction model. The three indexes are expressed by the following formulas:

$$Bias = \frac{1}{N} \sum_{i=1}^{N} |\hat{y}_i - y_i| \tag{10}$$

$$RMSE = \sqrt{\frac{\sum_{i=1}^{N}(y_i - \hat{y}_i)^2}{N}} \tag{11}$$

$$R^2 = 1 - \frac{\sum_{i=1}^{N}(y_i - \hat{y}_i)^2}{\sum_{i=1}^{N}(y_i - \overline{y})^2} \tag{12}$$

where $\hat{y}_i$ represents the predicted value, $y_i$ stands for the actual value, *N* is the number of samples, and $\overline{y} = \frac{1}{N} \sum_{i=1}^{N} y_i$.

## 4. Results and Discussion

### 4.1. Experimental Configuration and Model Training

This experiment was based on the PyTorch1.7.1 deep learning framework. The desktop computer used in the experiment was equipped with the Windows 11 operating system and Intel (R) Core (TM) i5-12490f CPU 3.00 GHz processor, and the running memory was 16 GB. Considering the GPU demand for load calculation, the GPU with the NVIDIA GeForce GTX 3060, 12 GB of video memory, and Python3.7 was selected, and other supporting software resources were CUDNN11.0 and PyCharm. In this experiment, the YOLOv7 model trained the crown detection and extraction model of *M. glyptostroboides* through transfer learning. The number of training iterations was set to 300, the batch_size of model training was set to 2, and the image input size was 640 × 640.

To better display the crown recognition and crown width measurement effects of the different models, the orthograph of the whole plot was inputted into the model for recognition and extraction. Given its size was too large, the orthograph could not be directly identified by the model. After reading, the orthograph was clipped into several small pictures via a program, with each small picture three times the estimated crown width, and 1.5 was selected as the overlapping rate to identify the divided small pictures. Meanwhile, the crown that was repeatedly identified in the overlapping area was deleted, and then the predicted frame was zoomed, followed by the offset calculation, and the final identified crown coordinates were obtained. Each identified crown position was marked with a red detection frame, and the crown width was calculated through the position of the detection frames and the number of pixels. The actual width of the crown was selected by using the labeling tool, and then the number of pixels for the crown length and width was calculated by using the program to extract the coordinates of the selection box. After extracting the number of pixels for the length and width, the real crown width was solved according to the proportion. The identification effect on the orthograph is shown in Figure 14.

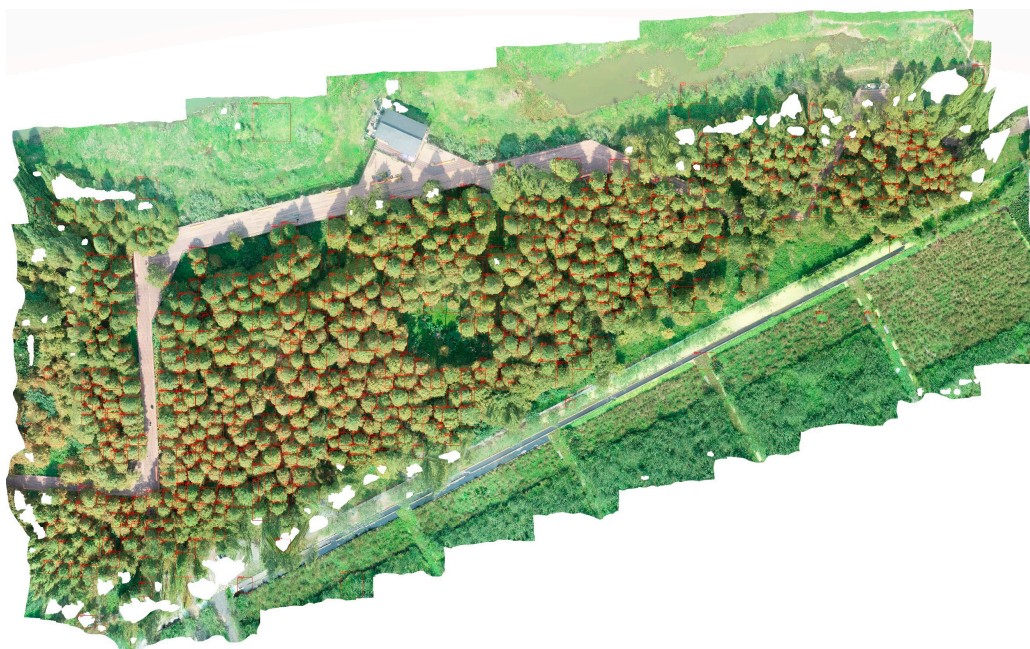

**Figure 14.** Identification effect of the model on the whole orthograph.

To verify the effectiveness of the improved model and the role of SimAM and SIoU in different tasks, ablation experiments were set, respectively, for the two tasks. In the task of crown detection, different improved models were used to train and test *M. glyptostroboides* datasets, and such evaluation indexes as the precision, recall, F1-score and mAP@0.5 were compared. In the task of the crown width measurement, an untrained and untested area was chosen; the bias, RMSE, and $R^2$ were calculated through the crown width data, which were calculated by the model and manually measured crown width data. Figure 15 shows the identification diagrams of the four models.

Where M1, M2, M3, and M4 represent YOLOv7, YOLOv7+SimAM, YOLOv7+SIoU, and YOLOv7+SimAM+SIoU, respectively.

### 4.2. Comparison of Crown Detection Performance

Table 1 shows that the improved M4 made great progress in the crown identification task of *M. glyptostroboides*. On the premise that precision is similar to that of M1, the recall and F1-score grew by 8.93% and 5.68%, respectively, and mAP@0.5 reached 94.34%, which is the highest among the four models. Meanwhile, the precision of M2 was improved by the SimAM module, reaching 98.03%, and the mAP@0.5 was 92.99%, which is slightly lower than that of M4, indicating that the average accuracy of the model is poorer than that of M4. After the introduction of the C3 module, the parameter size of the model decreased by almost 10%, demonstrating the effectiveness of the C3 module as lightweight. Compared with the original model, all the indexes of the M3 improved with the SIoU decrease, indicating that the sole substitution with SIoU did not exert any effect on the task of crown detection. The ablation experiments show that the SimAM attention module effectively improved the indexes of the model in the crown detection task, and the C3 module greatly shortened the training time required by the model. Although the precision slightly decreased after combining the SIoU with SimAM, the other three indexes were improved to some extent compared with M2, verifying that the combination of SIoU and SimAM helped improve the overall identification effect of the model.

**Table 1.** Ablation experiments on YOLOv7 crown detection models improved by different means.

| Method | Improvement Method | Precision | Recall | F1-Score | mAP@0.5 | Parameter Size | Training Time |
| --- | --- | --- | --- | --- | --- | --- | --- |
| M1 | YOLOv7 | 96.87% | 82.14% | 88.90% | 89.34% | 7.30 MB | 1.92 h |
| M2 | YOLOv7+SimAM | 98.03% | 89.29% | 93.46% | 92.99% | 6.63 MB | 1.31 h |
| M3 | YOLOv7+SIoU | 86.78% | 82.14% | 84.40% | 85.30% | 7.31 MB | 1.96 h |
| M4 | YOLOv7+SimAM+SIoU | 96.23% | 91.07% | 94.58% | 94.34% | 6.63 MB | 1.42 h |

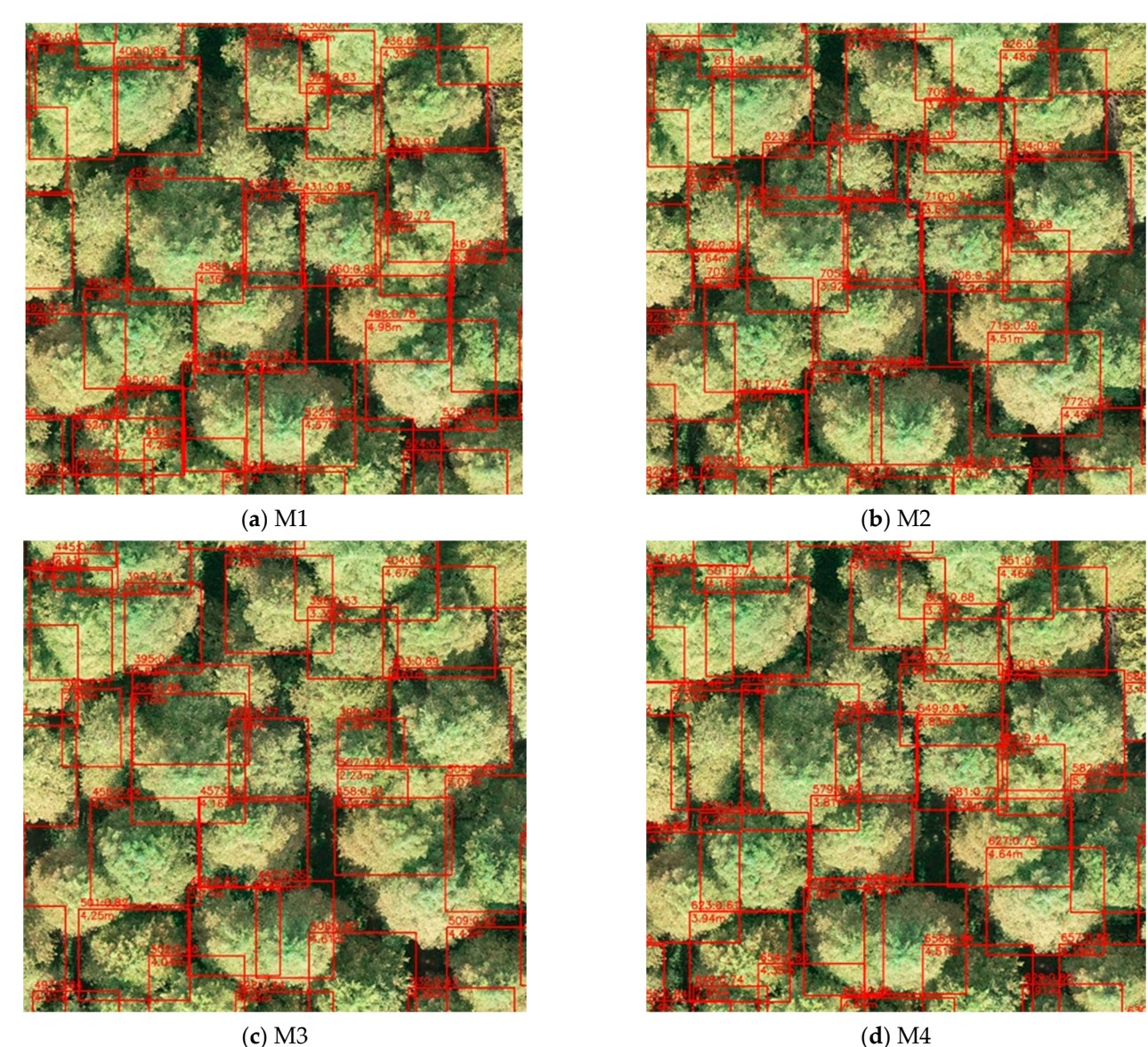

**Figure 15.** Four methods of model recognition renderings.

Figure 16 shows the line charts of the mAP@0.5 changes of the four models during training.

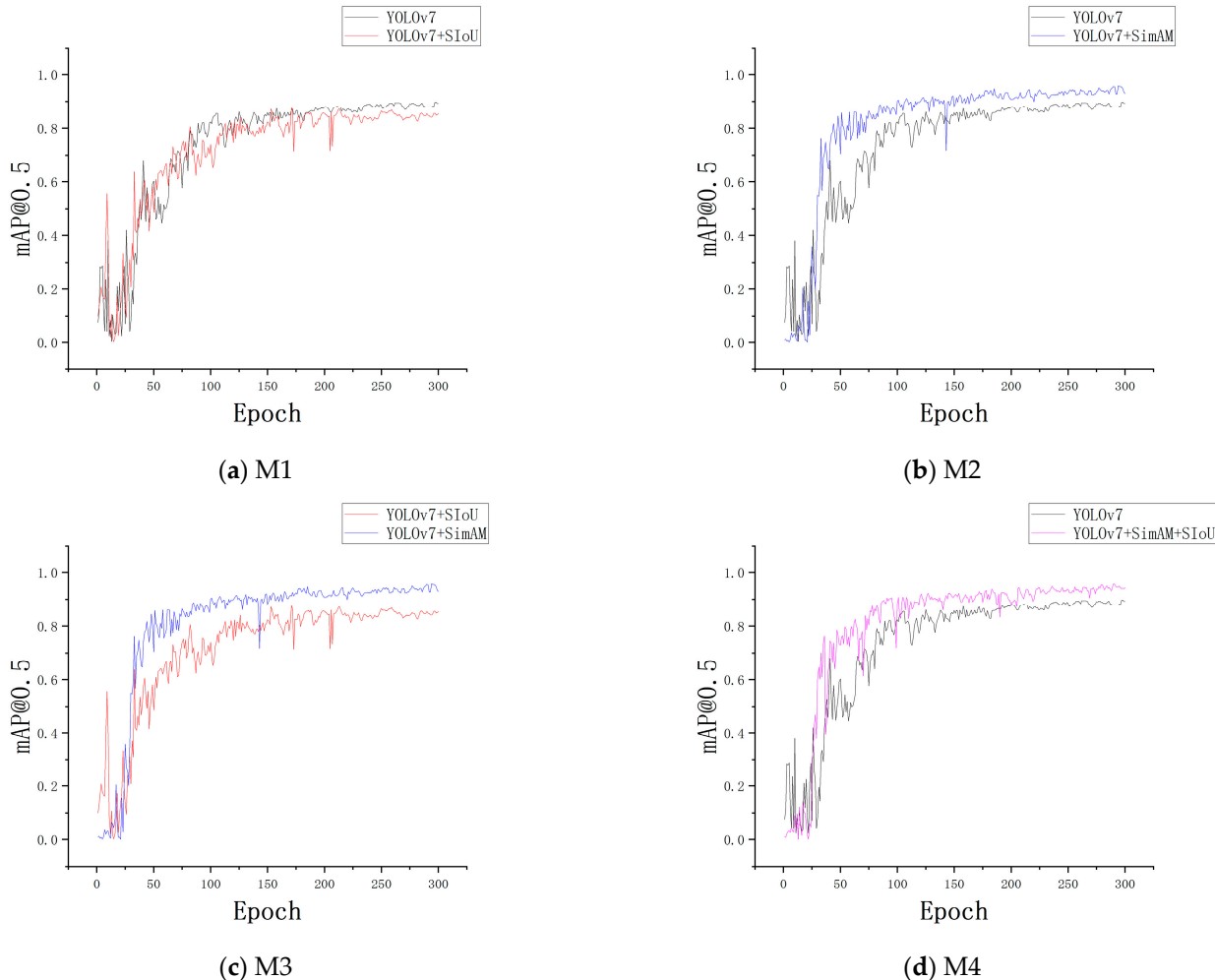

**Figure 16.** Line charts for comparison of mAP@0.5 changes of 4 models during training.

Figure 16 shows that, in the process of model training, the average accuracy of the model could be improved by SimAM; however, the average accuracy might decline to some extent if the model is improved by only using the SIoU. This phenomenon occurs because crown detection is different from the common target detection task. Under some circumstances, individual auxiliary branches of some trees will extend in the direction of strong light due to the light difference between forest lands and the complexity of forest environments. When growing up, these auxiliary branches will be mistakenly identified as a single crown. When the SIoU serves as the loss function, such auxiliary branches will be incorrectly identified as independent crowns, partly related to shape loss in the SIoU, thus decreasing the model indexes. However, when SimAM is introduced and combined with the advantages of the attention mechanism, the false identification in the case of the sole use of the SIoU is reduced, and independent crowns can be identified accurately. Thus, the overall effect of the model integrating SimAM and SIoU is good. Ablation experiments prove that, in the task of crown detection, SimAM helps to improve the model identification and average accuracies, and the overall model effect is good when SIoU and SimAM are combined.

### 4.3. Comparison of Crown Width Measurement Accuracy

In the task of crown width measurements, if the model identification frame accurately selects the crown frame and if the identification frame is close to the edge of the crown, this directly determines the accuracy of the crown width measurement. Figure 17 shows

the linear regression diagram of the estimated crown and actual crown widths of the four models in crown width measurement.

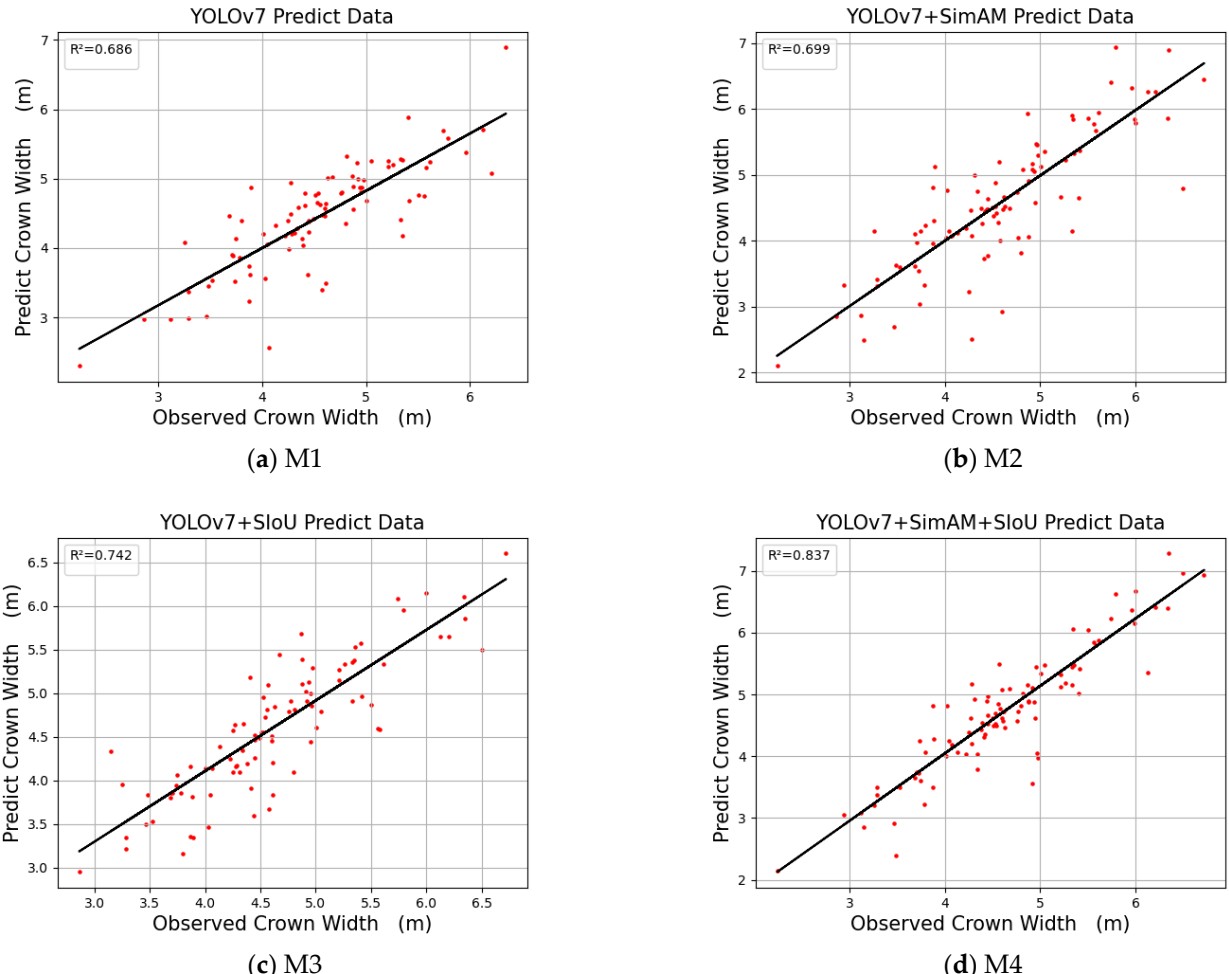

**Figure 17.** Linear regression diagram of crown width extraction accuracy of the 4 models.

Figure 17 shows that M4 achieved the highest crown width extraction accuracy. Table 2 shows no difference between M1 and M2 in $R^2$, while the $R^2$ values of M3 and M4 are 0.056 and 0.151 higher than M1, with an increase of 8.16% and 22.01%, respectively. These results reveal that the improved model made some progress in the task of crown width measurement. The ablation experiments show that the SimAM attention module hardly played a role in this task; however, after adding the SIoU module, the $R^2$ of the model obviously increased, indicating that adding the SIoU module helps improve the accuracy of the crown width measurement. After combining the advantages of the SimAM attention mechanism and the SIoU loss function, M4 achieved the best effect in the task of the crown width measurement, with $R^2$ reaching 0.837.

**Table 2.** Ablation experiments on crown width extraction effects of YOLOv7 models improved by different means.

| Method | Improvement Method | Bias | RMSE | $R^2$ |
|---|---|---|---|---|
| M1 | YOLOv7 | 0.322 | 0.457 | 0.686 |
| M2 | YOLOv7+SimAM | 0.394 | 0.554 | 0.699 |
| M3 | YOLOv7+SIoU | 0.305 | 0.412 | 0.742 |
| M4 | YOLOv7+SimAM+SIoU | 0.304 | 0.424 | 0.837 |

### 4.4. Comparison of Crown Identification and Crown Width Measurement Accuracy between Different Models

Table 3 shows that the precision and F1-score of the improved M4 model are higher than those of the two-stage detection model Faster-RCNN, reaching 96.23% and 91.07%, respectively; however, the recall (91.07%) is slightly lower than that of Faster-RCNN. However, all three indexes are higher than those of the SSD, which is the same single-stage detection model. Regarding the parameter size and inference time, the improved YOLOv7 has the smallest parameter size and the shortest inference time. Compared to the two-stage detector Faster-RCNN, the improved YOLOv7 parameter size and inference time are only 12.42% and 15.80% of Faster-RCNN, respectively, fully proving the effectiveness of the improved method in improving the accuracy of the original YOLOv7 model. Thus, the single-stage detection model YOLOv7 surpasses the two-stage detection model Faster-RCNN in terms of crown identification.

**Table 3.** Ablation experiments on crown width extraction effects of YOLOv7 models improved by different means.

| Method | Improvement Method | Precision | Recall | F1-Score | mAP@0.5 | Parameter Size | Inference Time |
|--------|--------------------|-----------|--------|----------|---------|----------------|----------------|
| M4 | YOLOv7+SimAM+SIoU | 96.23% | 91.07% | 94.58% | 94.34% | 6.63 MB | 18.2 ms |
| M5 | SSD | 94.90% | 72.50% | 82.20% | 86.40% | 9.28 MB | 62.4 ms |
| M6 | Faster-RCNN | 90.19% | 92.00% | 91.09% | 96.21% | 53.40 MB | 115.2 ms |

Figure 18 shows the canopy detection results of the three target detection models.

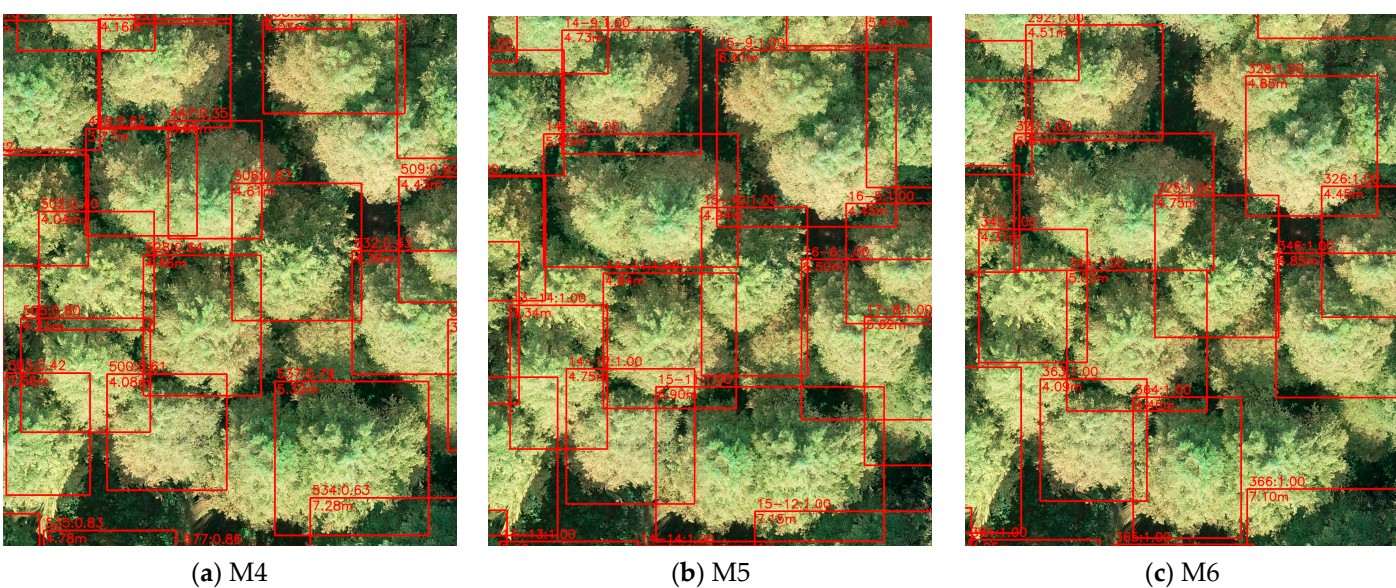

(**a**) M4          (**b**) M5          (**c**) M6

**Figure 18.** (**a**) Identification effect picture of YOLOv7 model improved by SimAM and SioU; (**b**) identification effect picture of SSD model; and (**c**) identification effect picture of Faster-RCNN model.

Figure 19 displays the linear regression diagram of the estimated crown and actual crown widths of three different models.

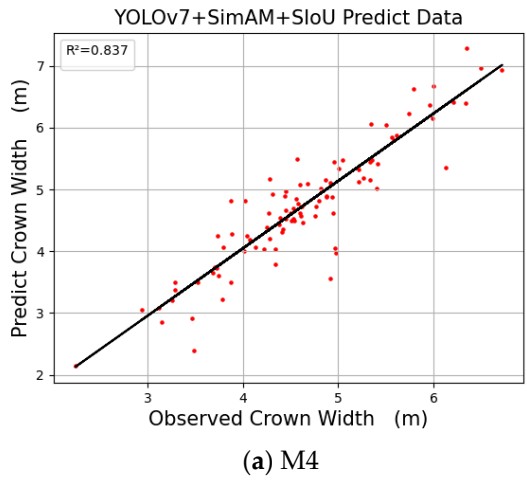

(**a**) M4

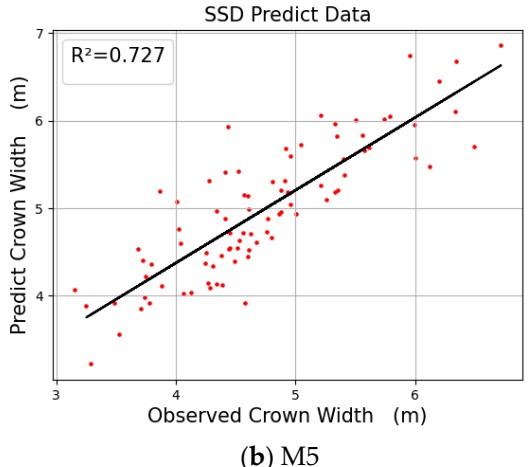

(**b**) M5

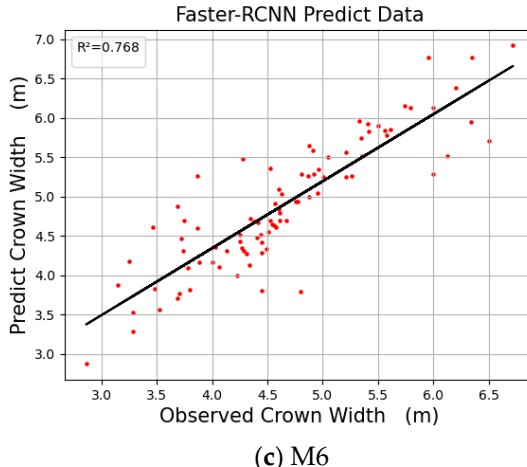

(**c**) M6

**Figure 19.** Linear regression diagram of crown width extraction accuracy of the three models.

As can be seen in Figure 19 and Table 4, the improved YOLOv7 model exhibits a strong crown extraction ability on the *M. glyptostroboides* forest than the other detectors. By using SimAM and SIoU to improve the single-stage detector YOLOv7, the RMSE reached 0.424 in the *M. glyptostroboides* forest with a complex background, which is 0.063 lower than that of the SSD, indicating that YOLOv7 works better than the single-stage detector SSD. Compared with the two-stage detector Faster-RCNN, the $R^2$ of M4 was 0.837, which is 0.11 higher than that of the single-stage detector SSD and 0.069 higher than that of the two-stage detector Faster-RCNN, fully demonstrating the feasibility of the improvement method and the superiority of the improved YOLOv7 model in crown identification and crown width extraction of *M. glyptostroboides* forests.

**Table 4.** Evaluation of crown width extraction effect indexes of different models.

| Method | Improvement Method | Bias | RMSE | $R^2$ |
|--------|--------------------|------|------|-------|
| M4 | YOLOv7+SimAM+SIoU | 0.304 | 0.424 | 0.837 |
| M5 | SSD | 0.362 | 0.487 | 0.727 |
| M6 | Faster-RCNN | 0.356 | 0.476 | 0.768 |

**5. Conclusions**

In this study, the *M. glyptostroboides* forest in Qingshan Lake in Lin'an District, Hangzhou City, Zhejiang Province, was photographed by UAVs, and a high-resolution orthograph was generated as the data source. The SimAM and SIoU modules were used to improve the single-stage model YOLOv7. The ablation experiment revealed that the SimAM and

SIoU improved the crown identification and crown width extraction effects, and the comprehensive effect was the best when the model was improved simultaneously by the two modules. Finally, the improved model that was based on the SimAM and SIoU achieved 94.34% of mAP@0.5 and 0.837 of $R^2$ in crown identification and crown width measurements, respectively, which are 5% and 0.151 higher than those of the original YOLOv7 model, respectively. The training time of the improved model is 26% shorter than that of the original model. Meanwhile, the improved model is superior to the single-stage detector SSD and two-stage detector Faster-RCNN in crown identification and width measurements. Compared with other target detection models, the improved YOLOv7 has a smaller parameter size and shorter inference time, sufficiently proving the feasibility of this improved method. *Metasequoia* usually grows near foothills or valleys where the terrain is gentle, deep, moist, or slightly waterlogged. Under such flat terrain, the UAV can ensure a constant flight height relative to the ground during aerial photography—a prerequisite for accurate crown width measurement. Based on the growth environment and topography of *Metasequoia*, the method of this study can also be extended to other artificial and partially natural *M. glyptostroboides* forests. The superiority of the YOLOv7 model in these two tasks provides selectivity for forestry staff in related works.

**Author Contributions:** Conceptualization, X.L. and C.D.; formal analysis, X.L., C.D., C.C., J.J., I.-K.H. and L.Y.; funding acquisition, Y.W. and C.D.; methodology, H.X., S.C. and S.H.; resources, X.L. and Y.W.; writing—original draft, C.C. All authors have read and agreed to the published version of the manuscript.

**Funding:** This research was partly funded by the ETPPRP and the McIntire Stennis program, and the Zhejiang Natural Science Foundation Project (grant number LQ21C160018).

**Data Availability Statement:** Data not applicable.

**Conflicts of Interest:** The authors declare no conflict of interest.

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
