# Peer review of "Crown Width Extraction of Metasequoia glyptostroboides Using Improved YOLOv7 Based on UAV Images"

_drones, doi:10.3390/drones7060336_

Round 1

Reviewer 1 Report

The study introduces the YOLOv7 by incorporating existing SimAM and SIoU loss function modules to detect tree crown in UVA. It is worth noting that a similar approach was previously used in an article titled "Measuring loblolly pine crowns with drone imagery through deep learning," published in the Journal of Forestry Research in 2022, which employed YOLOv3 for crown width extraction tasks. The difference is that this paper modified YOLOv7 by adding the existing SimAM and SIoU loss function modules, which are pre-existing algorithms. In my opinion, it is not suitable for publication as an academic paper though it is well-written and easy to understand.

The details of comments are as follows.

1) Introduction and bibliographic references:

The number of references is not sufficient. Some more recent contributions are, for example:

-Wu, Jintao, et al. "Extracting apple tree crown information from remote imagery using deep learning." Computers and electronics in agriculture 174 (2020): 105504.

-Lou, Xiongwei, et al. "Measuring loblolly pine crowns with drone imagery through deep learning." Journal of Forestry Research (2022): 1-12.

2) Description of the original algorithm YOLOv7:

-Different colored CBS have different settings, which are not explained in the paper.

-Some modules in Fig. 6 have issues with their input and output, such as the output of the last module "ELAN" in the backbone, as well as the input of SPPCSPC, Concat, and MP.

3) Description of the proposed SimAM and SIoU:

-This section lacks novelty as it simply adds existing modules and loss functions to YOLOv7.

-There is no specific description of the energy function "E" in SimAM.

-There is a derivation error in eq.2 in SIoU.

4) Description of the improved YOLOv7 network model:

- There are similar issues in the input and output of certain modules in both Fig. 12 and Fig. 6.

-C3, as a module in YOLOv5, is not specifically introduced in the paper.

5) Experimental section:

-The comparison of detection results in Fig. 14 is not prominent enough, it would be better to highlight the improvement.

-In section 4.3, the compared object detection algorithms are outdated.

-Further experiments should be conducted to compare the model parameter size and inference speed, to highlight the advantages of replacing the C3 module.

well-written and easy to understand

Author Response

尊敬的审稿人:

请参阅附件。

Reviewer 2 Report

This paper shows how to use Yolo based models for tree crown estimation. While this is interesting from the modeling and agricultural perspectives, it has nothing to do with drones. The UAV used in the study is a mere sensor but no specific challenges relate to the UAV are described.

The study itself is well presented does however lack a clear discussion of other models that tried to achieve a similar goal. Some papers are referenced but it remains unclear how the approached are different from the current one and what performance differences are there. This discussion should be enhanced and the results put into perspective.

In addition, the study is presented for one test plot only that looks like a rather young planted forest. Whether the approach would generalize to other locations remains unclear and is not discussed in the paper.

There are some English grammar and spelling mistakes. The abstract is affected too. Please revise your text carefully. 

Author Response

Dear Reviewer:

Reviewer 3 Report

This paper proposes an improved YOLOv7 model for the extraction of crown width from Metasequoia glyptostroboides. For this, the simple and parameterless attention model (SimAM) attention and SIoU modules have been integrated with YOLOv7. The experimental results show that the improved model achieves performance exceeding YOLOv7 in the task of crown detection and crown width measurement. This confirms the effectiveness of the proposed approach.

Accept in present form

Author Response

Dear Reviewer:

Round 2

Reviewer 1 Report

The paper is well-written and comprehensible. The research topic, exploring the development trend of this field, is highly significant. The description of the research ideas is relatively clear, and the experimental comparison work is relatively comprehensive.